# The Potential of Recycling the High-Zinc Fraction of Upgraded BF Sludge to the Desulfurization Plant and Basic Oxygen Furnace

**Anton Andersson** [1,*], **Mats Andersson** [2], **Elsayed Mousa** [3,4], **Adeline Kullerstedt** [3], **Hesham Ahmed** [1,4] , **Bo Björkman** [1] and **Lena Sundqvist-Ökvist** [1,3]

1   Department of Civil, Environmental and Natural Resources Engineering, Luleå University of Technology, 97187 Luleå, Sweden; hesham.ahmed@ltu.se (H.A.); bo.bjorkman@ltu.se (B.B.); lena.sundqvist-oqvist@ltu.se (L.S.-Ö.)
2   SSAB Europe, 97437 Luleå, Sweden; mats.andersson@ssab.com
3   Swerim AB, 97125 Luleå, Sweden; elsayed.mousa@swerim.se (E.M.); adeline.morcel@gmail.com (A.K.)
4   Central Metallurgical Research and Development Institute, Cairo 12422, Egypt
*   Correspondence: anton.andersson@ltu.se; Tel.: +46-920-493409

**Abstract:** In ore-based steelmaking, blast furnace (BF) dust is generally recycled to the BF via the sinter or cold-bonded briquettes and injection. In order to recycle the BF sludge to the BF, the sludge has to be upgraded, removing zinc. The literature reports cases of recycling the low-zinc fraction of upgraded BF sludge to the BF. However, research towards recycling of the high-zinc fraction of BF sludge within the ore-based steel plant is limited. In the present paper, the high-zinc fraction of tornado-treated BF sludge was incorporated in self-reducing cold-bonded briquettes and pellets. Each type of agglomerate was individually subjected to technical-scale smelting reduction experiments aiming to study the feasibility of recycling in-plant residues to the hot metal (HM) desulfurization (deS) plant. The endothermic reactions within the briquettes decreased the heating and reduction rate leaving the briquettes unreduced and unmelted. The pellets were completely reduced within eight minutes of contact with HM but still showed melt-in problems. Cold-bonded briquettes, without BF sludge, were charged in industrial-scale trials to study the recycling potential to the HM deS plant and basic oxygen furnace (BOF). The trials illustrated a potential for the complete recycling of the high-zinc fraction of BF sludge. However, further studies were identified to be required to verify these results.

**Keywords:** recycling; blast furnace sludge; smelting reduction; desulfurization; basic oxygen furnace; cold-bonded briquettes; cold-bonded pellets; low-sulfur binders

## 1. Introduction

Ore-based steelmaking generates a variety of residues including dusts, sludges, scales, and slags. Recycling of these residues within the process or via other applications is essential for the sustainable production of steel. Domestic environmental legislation [1] as well as the cost of raw material and energy continue to drive efforts towards increased recycling. However, the recycling has to be economically feasible and possible from a process technical standpoint.

One of the residues that is in general recycled completely—with no landfill—is the dry blast furnace (BF) dust. The BF dust is collected in the gas-cleaning equipment treating the top-gas from the BF. In addition to the coarse and dry dust, a wet finer residue is normally collected by scrubbing the gas; namely, the BF sludge. The BF sludge is generally landfilled despite having a chemical composition dominated by iron and carbon. There are three principal reasons that this residue is not recycled:

the zinc content, the fine particle size distribution and the water content. Generally, the zinc content is the limiting factor. Depending on how the BF is operated, the top-gas is more or less the main outlet of zinc from the furnace. Therefore, if the dust is recycled to the BF, the sludge needs to be pre-treated, removing zinc, before recycling in order to avoid its accumulation. The effects of zinc in the BF include increased reductant rates, reduced life of carbon-based refractories and scaffold formations, which may lead to a disturbed descent [2]. Thus, avoiding the accumulation of zinc in the furnace is essential.

The removal of zinc from BF sludge and recycling of the low-zinc fraction via the sinter [3–7] or cold-bonded pellets [7] to the BF has been implemented in industrial scale. However, on-site recycling of the high-zinc fraction generated in the dezincing process has not been reported. A logical step in recycling of the high-zinc fraction is to consider the available on-site processes, e.g., the hot metal (HM) desulfurization (deS) plant and basic oxygen furnace (BOF).

The BOF has been acknowledged as an alternative route for recycling of off-gas dusts from the integrated steel plant [8–12]. In one publication, recycling to the BOF by replacing the sinter used as coolant was stated to be limited in tonnage and was recognized as a partial solution for recycling of integrated steel plant dusts [8]. Nonetheless, industrial-scale trials have shown that off-gas dust from the BOF as well as BF dust can successfully be recycled via cold-bonded agglomerates to the BOF in the amounts of 23 [9] and 40 [10] kg per ton HM (kg/tHM). In the former study, 23 kg/tHM was the maximum recycling rate used in the trials [9]. In another study, cold-bonded briquettes were shown to be suitable to recycle all BOF sludge back to the BOF [11]. In addition, hot briquetting has been employed to recycle the BOF dust back to the BOF in industrial practice [12]. As the BOF has been recognized as a possible recycling route for in-plant residues, a logical succession is to study the potential of recycling the high-zinc fraction of upgraded BF sludge using this process.

Recycling to the BOF sets limitation on the agglomerates regarding their sulfur content, as recycling to this process is accompanied by sulfur pick-up in the crude steel [10]. This might limit the recycling rate in this process depending on the quality requirement of the produced steel and the sulfur removal capacity of the BOF process. Tang et al. [10] reported steel grades with sulfur requirements stricter than 0.008 wt.% were not eligible for recycling whereas steel grades allowing up to 0.015 wt.% sulfur could be used to recycle up to 40 kg/tHM. In order to meet the required quality of the steel, recycling prior to the deS of the HM should be considered as well. Prior to the current paper, research on this recycling route was yet to be reported.

In the present paper, smelting reduction of cold-bonded agglomerates in the form of briquettes and pellets was studied in technical-scale experiments aiming for recycling the high-zinc fraction of upgraded BF sludge in the HM deS plant. In addition, the potential for recycling the sludge was studied in industrial-scale trials by charging cold-bonded briquettes, without BF sludge, to the HM deS plant. In order to study the feasibility of improving the recycling capacity while maintaining the high quality of the final steel, the cold-bonded briquettes were charged to the BOF in industrial-scale trials as well. Furthermore, the sulfur pick-up of the crude steel was addressed by producing and characterizing cold-bonded briquettes produced with binders of low-sulfur contents in laboratory-scale.

## 2. Materials and Methods

### 2.1. Technical-Scale Experiments

#### 2.1.1. Material Characterization and Agglomeration

BF sludge from SSAB Oxelösund was upgraded, generating a low-zinc and high-zinc fraction, by utilizing the tornado process. The tornado is a high-velocity cyclone operating on pre-heated air. Tikka et al. [13] have previously described the equipment. The average iron, carbon, sulfur, and zinc content based on three samples of the high-zinc fraction of the BF sludge is presented in Table 1. The samples were taken during the operation of the tornado. A ThermoScientific ARL 9800 X-ray fluorescence (XRF) instrument with a rhodium tube (Thermo Fisher Scientific, Waltham, MA, USA) was used to determine the iron content of the sludge. A LECO CS444 combustion infrared detection

analyzer (LECO, St. Joseph, MI, USA) was used for the determination of carbon and sulfur. Inductively coupled plasma sector field mass spectrometry (ICP-SFMS) was used to analyze the zinc content. The digestion of the sample prior to the ICP-SFMS analysis was achieved by microwave-assisted dissolution in a mixture of nitric acid, hydrochloric acid, and hydrofluoric acid. The analysis was performed using a Thermo Finnigan Element 1 ICP-SFMS instrument (Thermo Fisher Scientific, Waltham, MA, USA).

**Table 1.** Wt.% of iron, carbon, sulfur and zinc in the high-zinc fraction of the upgraded blast furnace sludge.

| Fe | C | S | Zn |
|------|------|------|------|
| 29.6 | 19.5 | 0.31 | 1.57 |

The high-zinc fraction of the tornado-treated BF sludge was incorporated in both cold-bonded briquettes and pellets using the recipe presented in Table 2. Desulfurization scrap is fines of the magnetic fraction of the deS slag. Secondary dust is a dry filter dust collected from the steel shop. The recipe presented in Table 2 was designed to form self-reducing agglomerates.

**Table 2.** Recipe (wt.%) used for the briquettes and pellets in the technical-scale experiments.

| High-Zinc Fraction of BF Sludge | Desulfurization Scrap | Secondary Dust | Cement |
|---------------------------------|----------------------|----------------|--------|
| 25 | 50 | 15 | 10 |

Before the briquetting, the materials were mixed in a SoRoTo 40 L-33 (Soroto, Glostrup, Denmark) forced action mixer with several impellers. The briquetting was accomplished using a TEKSAM VU600/6 unit (TEKSAM, Hanstholm, Denmark) operating vibrating press technology. The hexagonal briquettes were approximately 7 cm high and 6.5 cm edge to edge. The briquettes were cured in humidified atmosphere for 24 h and then left in ambient room conditions for about 28 days. Prior to the experiments, the cured briquettes were dried to avoid explosions.

The deS scrap in the pellet recipe was ground using a rod mill to generate an appropriate particle size distribution for pelletization. The pellets were produced using a Mars Mineral DP14 Agglo Miser (Mars Mineral, Mars, PA, USA) equipped with a pelletizing disc of 35.6 cm in diameter. Screening of the pellets was performed to achieve a narrow size fraction between 9.5 mm and 10 mm. After curing, the pellets were dried to avoid explosions during the experiments.

2.1.2. Smelting Reduction Experiments

The smelting reduction experiments utilizing the briquettes were performed in an induction furnace with 80 kg of HM. A smaller induction furnace with 10 kg of HM was used in the experiments testing the pellets. In both cases, pig iron from BF No. 3 at SSAB Luleå was re-melted. The temperature of the melt during the experiments was aimed for 1350 °C. The principle of the tests was the same in both setups: the agglomerate was added to the surface of the melt and removed and quenched in nitrogen gas after predetermined times. The briquettes were scooped out of the melt whereas the pellets were tied with platinum wires and lifted out of the melt.

The mass loss during the smelting reduction was measured by recording the weight of each agglomerate before and after contact with the melt. The agglomerates were analyzed for the chemical composition using XRF, titration (ISO 9035) and LECO analysis. Furthermore, a PANalytical Empyrean X-ray diffraction (XRD) unit operating a cobalt tube (Malvern Panalytical, Almelo, The Netherlands) was used to determine the mineralogical composition.

## 2.2. Industrial-Scale Trials

Table 3 presents the recipe used to produce the briquettes for the industrial-scale trials. The fine fraction of steel scrap comes from the BOF process; it consists of material from the treatment of skulls and material from slopping during the blowing. In order to balance the water content of the mixture prior to briquetting, dry cast house dust from the BF and water were added. The chemical composition of the briquettes was determined using XRF and LECO analyses.

**Table 3.** Recipe (wt.%) used to produce the briquettes utilized in the industrial-scale trials.

| Fine Fraction of Steel Scrap | BOF Coarse Sludge | BOF Fine Sludge | Mill Scale from Cont. Casting | Cast House Dust | Cement | Water |
|---|---|---|---|---|---|---|
| 44 | 22 | 18 | 4 | 1 | 10 | 1 |

Prior to charging the briquettes to the HM deS plant, the briquettes were dried to 1.2 wt.% moisture to avoid incidents of smaller explosions. The briquettes were added in ten different trials in amounts ranging from 0.7 to 2.3 kg/tHM, which corresponded to 100–300 kg per heat. The additions were made to a ladle holding small amounts of HM in the bottom. After adding the briquettes, HM from the torpedo car was tapped into the ladle. The melt-in was studied visually and the effect of the addition on the final steel quality was evaluated.

The charging of the dried briquettes to the BOF was made together with the steel scrap. Nine trials with an amount of briquettes ranging from 4.9 to 10.9 kg/tHM were performed. These charging rates corresponded to 600–1250 kg of briquettes per heat. The effect on the final steel quality was evaluated.

## 2.3. Low-Sulfur Binders in Cold-Bonded Briquettes

As the sulfur removal capacity in the BOF may be limited, the recycling rate can be improved if low-sulfur binders are employed in the cold-bonded agglomerates. Briquette recipes using three different alternative binders were designed in order to study the feasibility of producing agglomerates lower in sulfur content with adequate strength for handling and recycling. Quicklime (CaO), slaked lime (Ca(OH)$_2$), and a synthetic organic binder were used to produce briquettes in laboratory scale, see Table 4. In addition, cement was used to produce the corresponding recipe for cement-bonded briquettes to offer a reference in terms of strength.

**Table 4.** Recipes (wt.%) used to produce the briquettes with the low-sulfur binders.

| Binder | Fine Fraction of Steel Scrap | BOF Coarse Sludge | BOF Fine Sludge | Mill Scale from Cont. Casting | Binder |
|---|---|---|---|---|---|
| Cement [1] | 45.0 | 22.5 | 18.0 | 4.5 | 10.0 |
| Quicklime | 45.0 | 22.5 | 18.0 | 4.5 | 10.0 |
| Slaked lime | 45.0 | 22.5 | 18.0 | 4.5 | 10.0 |
| Synt. organic binder | 50.0 | 25.0 | 20.0 | 5.0 | 0.03 |

[1] Same recipe as in Table 3 (excluding cast house dust) but produced in technical scale in order to get a reference in terms of strength.

The strength of the briquettes was measured after the same curing procedure as described in Section 2.1.1. In addition, the strength of the briquettes produced using the low-sulfur binders was tested after curing for 72 h in CO$_2$ atmosphere (10 L/min) in a Nabertherm muffle furnace at a temperature of 50 °C. The strength was evaluated by drop tests on a metal plate from a height of 1.0 m. The number of consecutive drops was counted until the briquette was broken.

## 3. Results and Discussion

### 3.1. Technical-Scale Smelting Reduction Experiments

#### 3.1.1. Characterization of the Agglomerates

The chemical composition of the cold-bonded agglomerates is presented in Table 5. A representative subsample of a crushed and finely ground briquette was used in the analysis. The high calcium to silicon ratio is suitable for charging to the HM deS plant as it will not act detrimental for the sulfur removal capacity of the slag in the deS process.

**Table 5.** Chemical composition in wt.% of the cold-bonded agglomerates used in the technical-scale experiments.

| Fe Met. | Fe(II) | Fe(III) [1] | O in FeO$_x$ | CaO | SiO$_2$ | MgO | Al$_2$O$_3$ |
|---|---|---|---|---|---|---|---|
| 21.0 | 2.5 | 13.3 | 6.4 | 23.0 | 5.8 | 2.5 | 2.2 |
| **MnO** | **K$_2$O** | **TiO$_2$** | **V$_2$O$_5$** | **P$_2$O$_5$** | **Na$_2$O** | **Zn** | **C** |
| 1.9 | 0.19 | 0.68 | 0.43 | 0.09 | 0.38 | 0.93 | 11.6 |

[1] Calculated based on Fe$_{tot}$, Fe$_{met.}$, and Fe(II).

The mineralogical composition of the cold-bonded agglomerates is illustrated in Figure 1. The identified phases were: hematite (Fe$_2$O$_3$), magnetite (Fe$_3$O$_4$), iron (Fe), periclase (MgO), graphite (C), calcite (CaCO$_3$), and portlandite (Ca(OH)$_2$). Wüstite (FeO) was not present above the detection limit of the XRD. Thus, the Fe(II) content presented in Table 5 corresponds to the Fe(II) in magnetite. Furthermore, lime was not detected in the XRD suggesting that the calcium detected in the XRF was present as calcite and portlandite.

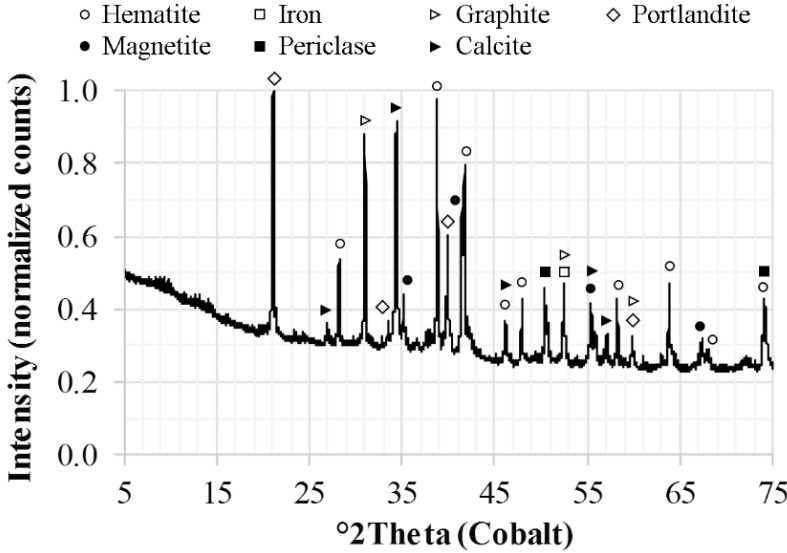

**Figure 1.** Diffractogram of the cold-bonded agglomerate used in the technical-scale experiments.

The reduction during smelting reduction of carbon-composite pellets has been proposed to occur mainly via the carbon inside the pellet, with little contribution from the carbon in the HM [14]. The carbon in the cold-bonded agglomerates of the present study is distributed as carbon in coke, coal, calcite, and carburized metallic iron. The metallic iron content of the agglomerate originates from the deS scrap. Therefore, this iron can be assumed to be carburized to 4.5 wt.% carbon. This means that 9.5% of the total carbon content in the agglomerate is found in this carburized metallic iron. Furthermore, making the conservative assumption that all calcium in the agglomerate is distributed as

calcite implies that 42.3% of the total carbon content will leave as carbon dioxide during the calcination upon heating, (Equation (1)).

$$CaCO_3 = CaO + CO_2,  \quad (1)$$

This means that the carbon content available for reduction is at least 48.2% of the total carbon content in the agglomerate. Therefore, the carbon to oxygen molar ratio is at least 1.2 when considering the moles of oxygen bound to iron (see Table 5). Thus, the agglomerate is self-reducible and the high C/O molar ratio is beneficial as increasing carbon to oxygen quotients have been shown to facilitate a faster reduction rate in self-reducing agglomerates [15–18].

### 3.1.2. Smelting Reduction of Cold-Bonded Briquettes

In the industrial process, the cold-bonded agglomerates would be charged to a ladle holding small amounts of HM. This HM is remaining desulfurized HM left over in the ladle when charging the BOF. The temperature of this HM was estimated to be 1350 °C. Thus, the temperature of the HM during the experiments in the 80 kg induction furnace was aimed for 1350 °C. Furthermore, in the industrial process, the time between charging the briquettes to the ladle, tapping HM from the torpedo car to the ladle and transporting the ladle to the HM deS plant is approximately ten minutes. Therefore, ten minutes was chosen as the longest time the briquettes were in contact with the melt in the technical-scale experiments. The propagation of the melt-in of the briquettes during these experiments is presented in Figure 2. A majority of the briquette is still to be melted after ten minutes.

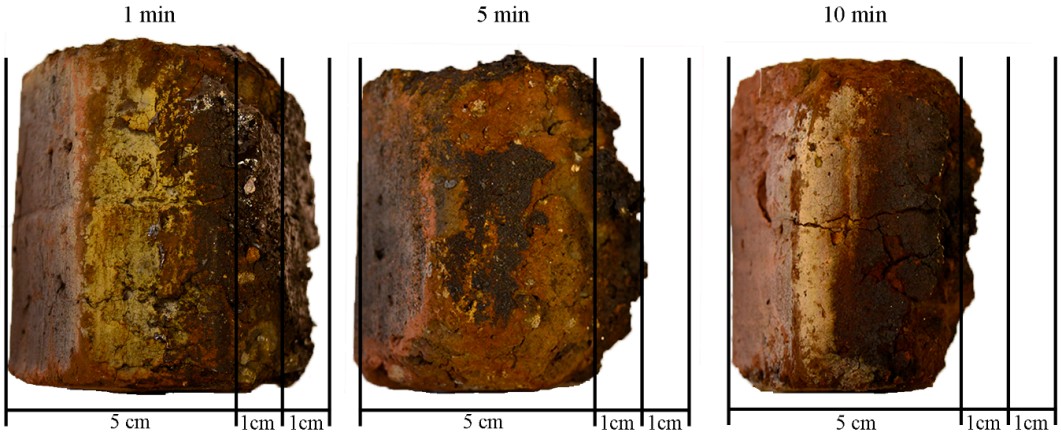

**Figure 2.** Propagation of the melt-in of the briquettes after 1, 5, and 10 min.

The results presented in Figure 3 suggest that the average carbon content for the whole briquette decreased slightly during the smelting reduction. Furthermore, the results of the valences of iron, determined by XRF and titration, indicated that the briquettes are not on average more reduced after being in contact with the melt for up to ten minutes as compared to the shorter times. The XRD analysis showed that the qualitative change in the mineralogy was limited to the detection of three peaks of brownmillerite ($Ca_2(Fe,Al)_2O_5$) in the briquettes having been in contact with the melt for seven minutes or longer. Furthermore, the diffractograms showed a distinct decrease in the relative intensities of the peaks corresponding to portlandite already after one minute in contact with the melt.

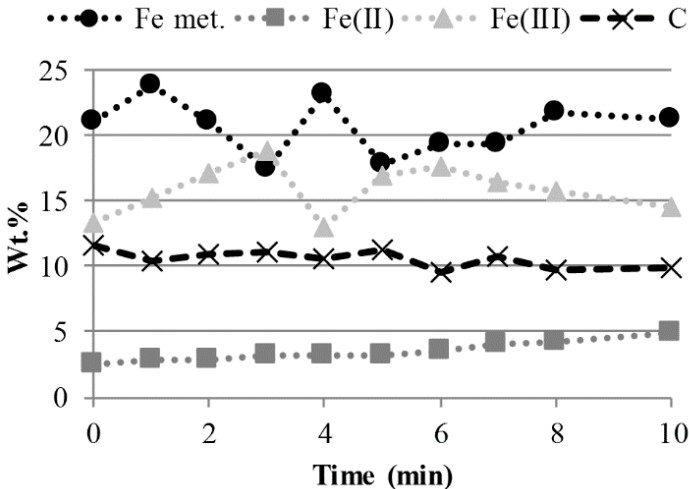

**Figure 3.** Wt.% of the carbon and the iron of different valences analyzed in samples of the briquettes after contact with the HM.

In order to study the propagation of the reduction in detail, XRD was run on samples from within the briquette that had been in contact with the melt for six minutes. The sample positions are presented in Figure 4a. Figure 4b presents the diffractograms of the *Melt*, *Middle* and *Atmosphere* samples. In the sample labeled *Melt*, the highest relative intensity was analyzed for metallic iron. In addition, wüstite was detected. The absence of wüstite in the diffractogram presented in Figure 1 suggests that this phase is only present as a reaction product. Furthermore, relative intensities of the higher iron oxides in the sample labeled *Melt* was low, suggesting a high degree of reduction. Moving towards the center of the briquette, the iron oxide analyzed for the highest relative intensity in the sample labeled *Melt-middle* was magnetite. Furthermore, wüsite was detected in this sample as well. In both the *Middle* and *Middle-atm.* samples, hematite was detected as the major iron phase and wüstite was not found. These results suggest that, as the reduction progress, the reduced part is melted and enters the HM, leaving the center part less reduced. However, the heat surrounding the rim of the briquette allowed for reduction of hematite as magnetite was analyzed for the highest relative intensity of the iron oxide phases in the sample labeled *Atmosphere*.

Wang et al. [19] studied the reduction-melting behavior of carbon composite pellets. In the study, four stages were identified to occur in the formation of an iron nugget separated from the slag phase: (i) heating, (ii) reduction, (iii) carburization of the iron, and (iv) melting of the carburized iron and slag separation [19]. Based on Figure 2, part of the briquette had gone through all the stages. Considering that the reduction during smelting reduction mainly occurs via the carbon inside the agglomerate [14], the cold-bonded briquette undergoes several endothermic reactions. These reactions include the reduction via carbon as well as the calcination reaction (Equation (1)) and dehydration of portlandite (Equation (2)). The dehydration of portlandite in nitrogen atmosphere has been shown to occur in temperatures between 355 and 442 °C [20]. In addition, direct carbonation of portlandite (Equation (3)) has been reported in temperatures between 200 and 355 °C in carbon dioxide atmosphere [20]. Thus, the high relative intensity of portlandite in the diffractogram of the sample labeled *Middle* in Figure 4 suggests that the middle part of the briquette was still undergoing the first stage, i.e., heating.

$$Ca(OH)_2 = CaO + H_2O, \quad (2)$$

$$Ca(OH)_2 + CO_2 = CaCO_3 + H_2O \quad (3)$$

Considering the low temperature of the center part of the briquette and the poor reduction and melt-in behavior, the idea of utilizing the same agglomerate recipe in cold-bonded pellets was to allow these smaller agglomerates to fully reduce and enter the melt.

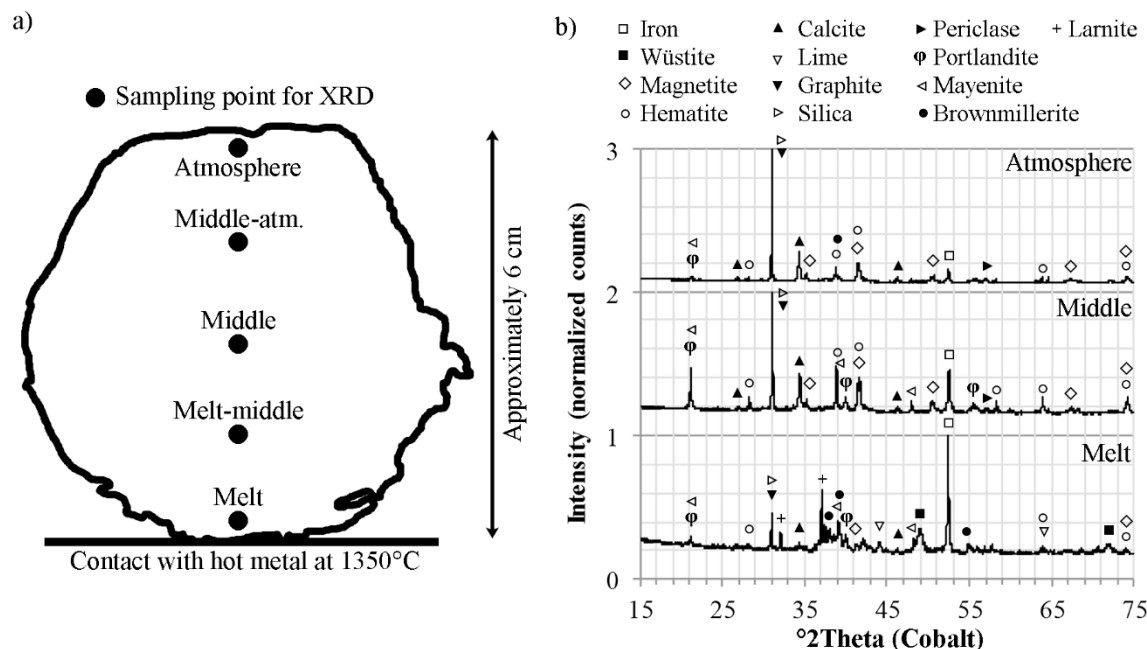

**Figure 4.** (**a**) Sample positions for X-ray diffraction (XRD) analyses of partly reduced briquette; (**b**) diffractograms corresponding to the sample positions.

### 3.1.3. Smelting Reduction of Cold-Bonded Pellets

The mineralogy of the pellets being in contact with the melt suggested that the iron oxides were reduced to amounts below the detection limit of the XRD after a time of contact between four and eight minutes, Figure 5a. However, although the pellets were completely reduced and smaller in size as compared to the briquettes, they still had melt-in problems, Figure 5b. These results are in line with the conclusions made by Ding and Warner [21] who found that the reduction of carbon-chromite composite pellets could be considerably faster than the dissolution when subjected to smelting reduction in high-carbon ferrochromium melts. Furthermore, they concluded that the rate-limiting step in the dissolution was the separation of the metallic phase from the slag phase. As an example, the dissolution time for a pellet of 10 mm in diameter could be lowered from >35 min down to 2–3 min by changing the melting point of the slag in the pellet [21]. In the present study, the start of the formation of the liquid slag phase was calculated to 1342 °C using the Equilib module of FactSage 7.2 with the FToxid database. This calculation was made by considering the $CaO$-$SiO_2$-$MgO$-$Al_2O_3$-$MnO$ system utilizing the slag composition based on Table 5. Although the liquid formation started at 1342 °C, the temperature of complete melting was calculated to 2204 °C. Thus, similar to the results of Ding and Warner [21], the thermodynamic calculations of the present study suggested that the rate of dissolution was limited by the high temperature of the melting interval of the slag phase.

Although the results of the technical-scale experiments suggested melt-in difficulties, the industrial-scale trials were considered of interest. The recipe of the agglomerate used in the industrial-scale trials was designed to have a lower melting interval of the slag phase as compared to the technical-scale experiments. Furthermore, the temperature of the HM charged from the torpedo to the ladle is generally 80 °C higher than the HM temperatures tested in the technical-scale experiments. These factors work towards an improved melt-in behavior of the agglomerate.

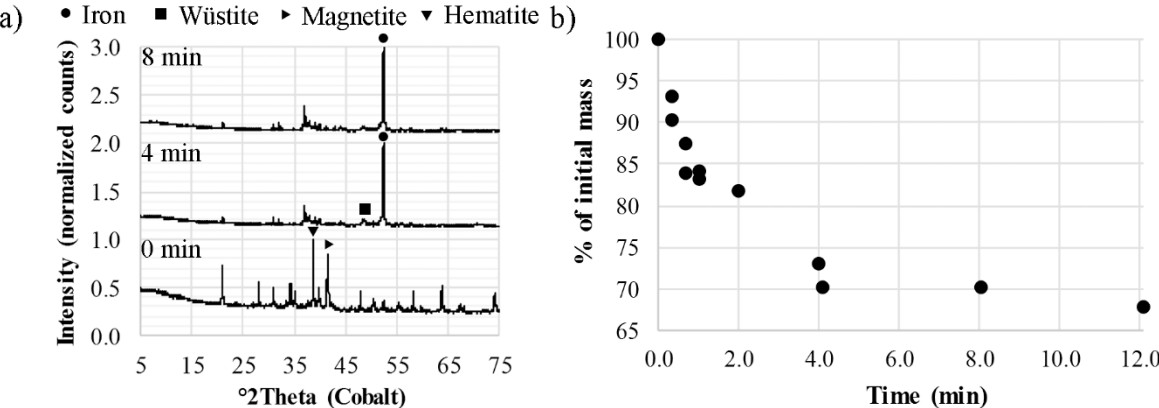

**Figure 5.** (**a**) Diffractograms of cold-bonded pellets after contact with HM; (**b**) mass loss curve during the smelting reduction of cold-bonded pellets.

## 3.2. Industrial-Scale Trials

### 3.2.1. Desulfurization Plant

The upgrading of BF sludge was not made in industrial-scale. Therefore, the recipe of the briquettes tested in the industrial-scale trials in the present study did not include BF sludge. Instead, they were designed to study the feasibility of the recycling route. If proven successful, the BOF sludge included in the recipes, Table 3, could be partially replaced with the high-zinc fraction of the upgraded BF sludge. Table 6 presents the chemical composition of the briquettes utilized in the industrial-scale trials. The calcium to silicon ratio was decreased, which lowered the calculated melting interval of the slag phase to 1329–1959 °C. Nonetheless, the calcium to silicon content in the briquette was still high and therefore suitable for charging prior to the deS as it will not deteriorate the sulfur removal capacity of the slag during the deS process.

**Table 6.** Chemical composition (wt.%) of the briquettes used in the industrial-scale trials.

| $Fe_{tot}$ | CaO | $SiO_2$ | MnO | $P_2O_5$ | $Al_2O_3$ | MgO | $V_2O_5$ | C | S |
|---|---|---|---|---|---|---|---|---|---|
| 42.3 | 22.6 | 8.8 | 1.8 | 0.2 | 3.1 | 4.4 | 1.3 | 2.0 | 0.3 |

Part of the iron in the briquettes is in the form of metallic iron coming from the steel scrap fines and the coarse and fine BOF sludge. The rest is iron oxides from both the coarse and fine BOF sludge as well as the mill scale. The degree of self-reducibility was not determined for the briquettes. However, considering the low carbon content, there is a risk that all iron oxides will not be reduced. In that case, the endothermic reduction reactions via carbon inside the briquette will not occur. Therefore, the rate of heating of the briquette will improve and the iron oxides may enter the slag phase prior to being reduced as proposed by dos Santos and Mourão [14]. This will lower the temperature of the start of melting and complete melting of the slag. Consequently, the reduction of the iron oxides will occur by the silicon in the HM [22] at the slag/hot-metal interface. Dos Santos and Mourão [14] suggested that carbon in the HM will participate in the reduction as well. In such a scenario, reducing the iron oxides in the slag phase is essential as these will control the oxygen partial pressure at the interface between the slag and HM, which lowers the sulfur partition ratio ($\%S_{slag}/\%S_{HM}$) and the rate of deS [22].

During the operation in the industrial-scale trial, the moisture content and strength of the briquettes allowed a safe operation without any incidents. Charging the briquettes to the ladle caused minor dusting. The melt-in of the briquettes prior to the start of the deS operation was evaluated visually. Charging up to 1.2 kg/tHM enabled melting of all added briquettes. However, only partial melt-in was noticed when charging 2.3 kg/tHM. Nonetheless, after the deS process, no briquettes were observed, indicating a successful melt-in.

Adding the briquettes to the ladle prior to the deS process did not affect the efficiency of the deS. The final steel quality was not impaired in any of the trials suggesting that up to 2.3 kg/tHM is possible to add in the process. This corresponds to 5400 metric tons of briquettes per year. This recycling rate will not result in the complete recycling of the high-zinc fraction of upgraded BF sludge at the present integrated steel plant. Therefore, recycling to the BOF was considered as well.

### 3.2.2. Basic Oxygen Furnace

Charging the briquettes together with the scrap to the BOF caused minor dusting. Nonetheless, the briquettes allowed for a safe operation. The dephosphorization, deP, expressed according to (Equation (4)) was improved when adding the cold-bonded briquettes together with the scrap, Figure 6a. The deP (trials) of Figure 6a represents the dephosphorization of the trials of the present study whereas the deP (average) represents the average dephosphorization of the corresponding steel type without briquette additions.

$$deP = \frac{\%P_{HM} - \%P_{CS}}{\%P_{HM}} \tag{4}$$

where $\%P_{HM}$ is the phosphorous content in HM and $\%P_{CS}$ is the phosphorous content in the crude steel.

The sulfur content of the briquettes caused sulfur pick-up in the crude steel, Figure 6b. The sulfur pick-up attributed to the briquettes was determined by material balance calculations where the analyzed or estimated sulfur contents of all ingoing and outgoing material streams were considered. The sulfur pick-up in the present study is comparable to that of Tang et al. [10] when considering the same range of kg of agglomerates added per ton HM to the BOF. Due to the sulfur pick-up in the crude steel, the charging of the cold-bonded briquettes to the BOF of the steel shop in the present study was limited to certain steel types. Based on this, the calculated annual capacity for recycling of briquettes in the BOF was determined to 8700 metric tons.

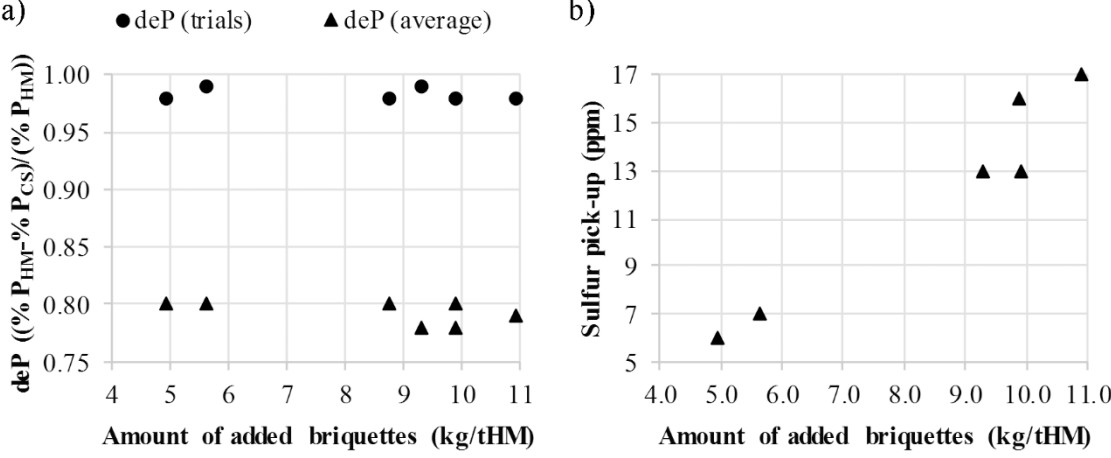

**Figure 6.** (**a**) Dephosphorization during the trials and the corresponding average for the steel code produced; (**b**) sulfur pick-up attributed to the briquettes calculated by mass balancing.

Based on the above, the total amount of briquettes that could be recycled to the steel shop, including both the HM deS plant and BOF, was estimated to 14,100 metric tons per year. Depending on which method is used for upgrading the BF sludge, the amount of solids reporting to the high-zinc fraction varies [23]. Consequently, this determines the required wt.% of the high-zinc fraction in the briquette recipe that results in the complete recycling of this fraction. Using hydrocycloning, 22 wt.% of the briquette recipe has to be constituted of the high-zinc fraction. The corresponding number for tornado-treated BF sludge is 46 wt.%. In the latter case, replacing both the coarse and fine BOF sludge in the briquette recipe, Table 3, would not result in the complete recycling of the high-zinc fraction. Therefore, in this scenario, means to improve the recycling rate of briquettes to the steel shop has to be considered. Furthermore, incorporating the high-zinc fraction of upgraded BF sludge

in the briquettes would introduce the endothermic self-reducing reactions. Consequently, the effect of the BF sludge additions on the melt-in has to be studied in order to verify the recycling potential. As agglomerates including high carbon-containing BF flue dust have been successfully recycled to the BOF [9,10], the rise of melt-in problems are most likely to be seen when recycling to the HM deS plant. Considering the results in the technical-scale experiments with pellets, the rise of melt-in problems could be tackled with reducing the agglomerate size and designing an appropriate melting temperature interval of the internal slag composition of the agglomerate.

*3.3. Low-Sulfur Binders in Cold-Bonded Briquettes*

When recycling to the BOF, the sulfur pick-up was the limiting factor in the present study. The sulfur content in the briquettes comes from the in-plant residues and the cement. The briquettes were bound with cement containing 1.37% sulfur. Thus, based on Table 6, 46% of the total sulfur content in the briquettes can be attributed to the cement. In order to study the feasibility of improving the recycling rate to the BOF, the use of binders with lower sulfur contents than cement was studied in laboratory-scale briquetting experiments.

As the briquettes were produced in limited amounts on the technical scale, the strength evaluation was done using drop tests instead of measuring the tumbling index via ISO 3271. The latter requires between 30 to 60 kg of material depending on the repeatability of the test. The strength of the cement-bonded briquettes was superior to the other alternatives, Table 7. The strength of the briquettes bound using the quicklime and slaked lime improved significantly after curing in $CO_2$ as compared to in ambient room conditions. Fernández Bertos et al. [24] reported that curing in $CO_2$ atmosphere activates cementitious compounds otherwise considered to have poor hydraulic properties. Furthermore, the carbonation process induced by the $CO_2$ generates carbonate cementation resulting in improved strength [24].

**Table 7.** Sulfur content of binders and number of drops before breaking after curing in ambient room conditions and $CO_2$, respectively.

| Binder | No. of Drops (Cured in Room) | No. of Drops (Cured in $CO_2$) | % S in Binder |
|---|---|---|---|
| Cement | 38 | / | 1.37 |
| Quicklime | 1 | 6 | 0.11 |
| Slaked lime | 1 | 5 | 0.53 |
| Synt. organic binder | 2 | 3 | 0.03 |

The "/" denotes that these were not cured in $CO_2$.

Robinson [25] characterized BOF coarse and fine sludge from the same steel shop as the present study. In both residues, portlandite was detected [25]. Thus, the improved strength observed after the curing in $CO_2$ of the briquette bound using the synthetic organic binder can be attributed to the portlandite content originating from the BOF coarse and fine sludge.

Using quicklime as a binder, replacing cement, decreased the sulfur content of the briquettes from 0.30 to 0.14%. By lowering the sulfur content of the briquettes to 0.14%, the recycling rate to the BOF can be increased. Recycling cement-bonded briquettes to the BOF at a rate of 8700 metric tons per year amounts to an annual sulfur load of 26 metric tons. Replacing cement by quicklime would allow for 18,600 metric tons of briquettes to be recycled each year while maintaining the same sulfur load. Thus, the total recycling rate to the steel shop would amount to 24,000 metric tons. Possibly, the strength requirement for handling and charging exceeds the strength measured for the $CO_2$-cured quicklime briquettes. Therefore, a realistic way forward would be to study other alternative binders or to lower the cement content in the original recipe, presented in Table 3.

## 4. Conclusions

In the present paper, the feasibility of recycling in-plant residues, focusing on the high-zinc fraction of upgraded BF sludge, to the HM deS plant and BOF was studied. The viability of recycling

the high-zinc fraction of tornado-treated BF sludge to the HM deS plant was studied in technical-scale smelting reduction experiments. The high-zinc fraction was incorporated in self-reducing cold-bonded briquettes and pellets. These agglomerates were placed in contact with HM for predetermined times finding that:

- The endothermic reactions occurring upon heating of the briquette lowered the heating and reduction rate, which hindered the melt-in.
- The pellets, being smaller than the briquettes, were fully reduced after eight minutes in contact with the HM but still showed melt-in problems.

The results of the technical-scale experiments were utilized to design a new cold-bonded briquette recipe, without the BF sludge fraction, that was charged in industrial-scale trials to the HM deS plant and BOF to study the recycling potential of the sludge:

- The HM deS plant could be used to recycle up to 2.3 kg of briquettes per tHM, which was the maximum amount tested in the study.
- Recycling briquettes together with the cooling scrap charged to the BOF enabled recycling of 10.9 kg of briquettes per tHM. The recycling rate was limited by the sulfur pick-up in the crude steel.

The annual recycling rate to the steel shop of the present study was estimated to 14,100 metric tons. Depending on the upgrading method used to treat the BF sludge, the study indicated that the high-zinc fraction could be completely recycled via these briquettes. However, due to the endothermic self-reduction associated with the BF sludge during the smelting reduction, further studies were identified to be required in order to verify the recycling potential. Furthermore, producing briquettes using low-sulfur binders in laboratory-scale experiments illustrated a potential of more than a two-fold increase in the recycling rate to the BOF. However, the strength of these agglomerates was considerably lower than that of cement-bonded briquettes.

**Author Contributions:** Conceptualization, B.B. and L.S.-Ö.; Methodology, A.A., A.K., M.A., and E.M.; Formal Analysis, A.A., M.A., and E.M.; Investigation, A.A., A.K., M.A., and E.M.; Writing—Original Draft Preparation, A.A.; Writing—Review & Editing, A.A., H.A., L.S.-Ö., and B.B.; Supervision, H.A., L.S.-Ö., and B.B.; Project Administration, H.A.; Funding Acquisition, L.S.-Ö. and B.B.

**Funding:** This research was funded by the Swedish Energy Agency and the research program Iron and Steel Industry Energy Use (JoSEn). The work was carried out within CAMM—Centre of Advanced Mining and Metallurgy at Luleå University of Technology.

**Conflicts of Interest:** The authors declare no conflict of interest.

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
