# Peer review of "The Potential of Recycling the High-Zinc Fraction of Upgraded BF Sludge to the Desulfurization Plant and Basic Oxygen Furnace"

_metals, doi:10.3390/met8121057_

Reviewer 1 Report

The article entitled "Recycling of the High-Zinc Fraction of Upgraded BF Sludge to the Desulfurization Plant and Basic Oxygen Furnace" is a well-written paper that aims studing the recycling of BF sludge containing high-zinc fraction. They clearly described the objectives. The procedure as well as the results are clearly presented in the document. High quality images are used with illustrative captions. Conclusions are supported in the results. And, they provide a significant number of relatively updated references.

Author Response

Thank you for reading the paper.

Reviewer 2 Report

“… agglomerates to the BOF in amounts of 23 [9] and 40 kg per ton hot metal (kg/tHM) [10].” should be “… agglomerates to the BOF in amounts of 23 kg [9] and 40 kg [10] per ton hot metal (kg/tHM).”

“…phase from the slag phase [21].” should be “…phase from the slag phase.”

Reference [4]

Honingh, S.; Van Weert, G.; Reuter M.A. TURNING BLAST FURNACE DUST INTO A SOURCE OF ZINC AND LEAD UNITS: A PROGRESS REPORT ON TESTWORK AT CORUS IJMUIDEN. In Proceedings of the Fourth Int. Symp. on Recycling of Metals and Engineered Materials, Pittsburgh, PA, USA, 22-25 October 2000; TMS: Pittsburgh, PA, USA, 2000.

Please put the title of the paper in lowercase letters

Same remark for the reference [16].

For the references, use rather “Abbreviated Journal Name” instead of “Full Journal Name”.

Author Response

Thank you for the input, we have addressed all your suggestions.

Author Response

Thank you for the many suggestions on how to improve the paper. Please see the attached file.

Round  2

Reviewer 3 Report

Please see the attached comments
